# CERTIFIED ROBUSTNESS TO CLEAN-LABEL POISONING USING DIFFUSION DENOISING

## ABSTRACT

We present a certified defense to clean-label poisoning attacks. These attacks work by injecting a small number of poisoning samples (e.g., 1%) that contain $p$-norm bounded adversarial perturbations into the training data to induce a targeted misclassification of a test-time input. Inspired by the adversarial robustness achieved by *randomized smoothing*, we show how an off-the-shelf diffusion model can sanitize the tampered training data. We extensively test our defense against seven clean-label poisoning attacks and reduce their attack success to 0–16% with only a negligible drop in the test time accuracy. We compare our defense with existing countermeasures against clean-label poisoning, showing that the defense reduces the attack success the most and offers the best model utility. Our results highlight the need for future work on developing stronger clean-label attacks and using our certified yet practical defense as a strong baseline to evaluate these attacks.

## 1 INTRODUCTION

A common practice in machine learning is to train models on a large corpus of data. This paradigm empowers many over-parameterized deep-learning models but also makes it challenging to retain the quality of data collected from various sources. This makes deep-learning-enabled systems vulnerable to *data poisoning* (Rubinstein et al., 2009; Nelson et al., 2008b; Biggio et al., 2012; Jagielski et al., 2018; Shafahi et al., 2018; Carlini, 2021)—where an adversary can alter a victim model's behaviors by injecting malicious samples (i.e., poisoning samples) into the training data.

But in practice, it may be challenging to make modifications to entire training data used for supervised classification, because they are often relatively small. As a result, recent work has developed *clean-label poisoning attacks* (Shafahi et al., 2018; Zhu et al., 2019; Aghakhani et al., 2021; Turner et al., 2019; Saha et al., 2020; Huang et al., 2020; Geiping et al., 2021b), where the attacker aims to control the victim model's behavior on a specific test input by injecting a few poisons that visually appear to be correctly-labeled, but in fact include human-invisible adversarial perturbations.

We present a defense against clean-label poisoning attacks inspired by defenses to test-time adversarial examples. Existing poisoning defenses fall into two categories *certified* and *heuristic*. Certified defenses offer provable guarantees, but often significantly decrease the utility of defended models at test-time, making them impractical (Ma et al., 2019; Levine & Feizi, 2020; Wang et al., 2022; Zhang et al., 2022). Heuristic approaches (Suciu et al., 2018; Peri et al., 2020; Hong et al., 2020; Geiping et al., 2021a; Liu et al., 2022) demonstrate their effectiveness against existing attacks in realistic scenarios. However, these defenses rely on unrealistic assumptions, such as the defender knowing the target test input (Suciu et al., 2018), or are evaluated against specific poisoning adversaries, leaving them ineffective against adaptive attacks or future adversaries.

**Our contributions.** *First*, we make two seemingly distant goals closer by presenting a certified defense against clean-label poisoning that *also* minimizes the decrease in clean accuracy. For any $p$-norm bounded adversarial perturbations to the training data, we ensure a certified accuracy higher than the prior certified defenses. The model trained on the tampered data classifies a subset of test input $x$ (or $x + \delta$, where $||\delta||_{\ell_p}$ in clean-label backdoor poisoning) correctly. To achieve this goal, we leverage the recent diffusion probabilistic diffusion models (Sohl-Dickstein et al., 2015; Ho et al., 2020; Nichol & Dhariwal, 2021). We use an off-the-shelf diffusion model to denoise the entire training data before training a model. In §3, we theoretically show how one achieves a certified accuracy under 2-norm bounded adversarial perturbation to the training data.

Removing adversarial perturbations in the training data before model training, we can decouple the certification process from the training algorithm. *Second*, we leverage this computational property our defense has and present a series of training techniques to alleviate the side-effects of employing certified defenses, i.e., the utility loss of a defended model. To our knowledge, we are the first certified defense that decouples these two processes. We minimize the utility loss by employing the warm-starting (Ash & Adams, 2020). We train a model a few epochs on the tampered data or initialize its model parameters using a pre-trained model. None of them alters the training process; thus, our defense's provable guarantee holds. In §4, we show they attribute to a better performance compared to existing certified defenses.

*Third*, we extensively evaluate our defense against seven clean-label poisoning attacks studied in two different scenarios under $\ell_2$- and $\ell_\infty$-norm bounds: transfer-learning and training a model from-scratch. In transfer-learning, our defense completely renders the attack ineffective with a negligible accuracy drop of 0.5% in the certified radius of 0.1 in 2-norm perturbations. We also reduce the attack success to 2–16% when a defender trains a model from scratch. We further compare our defense with six poisoning defenses in the prior work. We demonstrate more (or the same in a few cases) reduction in the attack success and less accuracy drop than those existing defenses.

We discuss research questions important in studying poisoning attacks and defenses but are far-neglected in the prior work. We suggest future work directions to answer them.

## 2 PRELIMINARIES ON CLEAN-LABEL POISONING

**Threat model.** A clean-label poisoning attacker causes a misclassification of a specific *target* test-time sample $(x_t, y_t)$ by compromising the training data $D_{tr}$ with poisoning samples $D_p$. If a victim trains a model $f$ on the poisoned training data $D_{ptr} = D_{tr} \cup D_p$, the resulting model $f_\theta^*$ is likely to misclassify the target instance to the adversarial class $y_{adv}$ while preserving the classification behavior of $f_\theta^*$ on the clean test-set $S$. The attacker crafts those *poisons* $(x_p, y_p)$ by first taking a few *base* samples in the same domain $(x_b, y_b)$ and then adding human-imperceptible perturbations $\delta$, carefully crafted by the attacker and also bound to $||\delta||_{\ell_p} \leq \epsilon$, to them while keeping their labels *clean* $(y_b = y_p)$. A typical choice of the bound is $\ell_\infty$ or $\ell_2$.

**Poisoning as a constrained bi-level optimization.** The process of crafting optimal poisoning samples $D_p^*$ can be formulated as the constrained bi-level optimization problem:

$$D_p^* = \arg\min_{D_p} \mathcal{L}_{adv}(x_t, y_t; f_\theta^*),$$

where $\mathcal{L}_{adv}(x_t, y_{adv}; f_\theta^*)$ is the adversarial loss function quantifying how accurately a model $f_\theta^*$, trained on the compromised training data, misclassifies a target sample $x_t$ into the class an adversary wants $y_{adv}$. $D_p$ is the set of poisoning samples we craft, and $D_p^*$ is the resulting optimal poisons.

While minimizing the crafting objective $\mathcal{L}_{adv}(x_t, y_t; f_\theta^*)$, the attacker also trains a model $f_\theta^*$ on the compromised training data, which is itself another optimization problem, formulated as follows:

$$f_\theta^* = \arg\min_\theta \mathcal{L}_{tr}(D_{ptr}, S; \theta),$$

where the typical choice of $\mathcal{L}_{tr}$ is the cross-entropy loss, and $S$ is the clean test-set. Combining both the equations becomes a bi-level optimization: find $D_p^*$ such that $\mathcal{L}_{adv}$ is minimized after training, while minimizing $\mathcal{L}_{tr}$ as well.

To make the attack inconspicuous, the attacker *constraints* this bi-level optimization by limiting the perturbation $\delta = x_p - x_b$ each poisoning sample can contain to $||\delta||_{\ell_p} < \epsilon$.

**Existing clean-label attacks.** Initial work (Shafahi et al., 2018; Zhu et al., 2019; Aghakhani et al., 2021) minimizes $\mathcal{L}_{adv}$ by crafting poisons that are close to the target in the latent representation space $g(\cdot)$. A typical choice of $g(\cdot)$ is the activation outputs from the penultimate layer of a pre-trained model $f(\cdot)$, publicly available to the attacker from the Internet. The attacks have shown effective in *transfer-learning* scenarios, where $f(\cdot)$ will be fine-tuned on the poisoned training data. The attacker chooses base samples from the target class $(x_b, y_{adv})$ and craft poisons $(x_p, y_{adv})$. During fine-tuning, $f(\cdot)$ learns to correctly classify poisons in the target's proximity in the latent representation space and classify the target into the class $y_{adv}$ the attacker wants.

Recent work focuses on making those poisoning attacks effective when $f$ is trained *from scratch* on the tampered training set. To do so, the attacker requires to approximate the gradients computed on $\mathcal{L}_{adv}$ that are likely to appear in any models. Huang et al. (2020) address this challenge by meta-learning; the poison-crafting process simulates all the possible initialization, intermediate models, and adversarial losses computed on those intermediate models. A follow-up work by Geiping et al. (2021b) alleviates the computational overhead of the prior work by proposing gradient matching, that aligns the gradients from poisoning samples with those computed on a target.

**Defenses against clean-label poisoning.** Early work on poisoning defenses focuses on filtering out poisons from the training data. Nelson et al. (2008a) and Suciu et al. (2018) compute the training samples negatively impacting the classification results on targets. But in practice, the defender does not know which test-time samples are the attack targets. Follow-up work (Rubinstein et al., 2009; Peri et al., 2020; Tran et al., 2018) leverages unique statistical properties (*e.g.*, spectral signatures) that can distinguish poisons from the clean data. All these approaches depend on the data they use to compute the properties. Recent work (Geiping et al., 2021a; Borgnia et al., 2021; Hong et al., 2020; Liu et al., 2022) thus reduces the dependency by proposing data- or model-agnostic defenses, adapting robust training or differentially-private training. While shown effective against existing attacks, those defenses are not *provable*.

A separate line of work addresses this issue by proposing *certifiable* defenses (Levine & Feizi, 2020; Wang et al., 2022; Weber et al., 2023), which guarantee the correct classification of test-time samples when an adversary compromises the training data with the $k$ poisons whose perturbation radius of $r$. Levine & Feizi (2020); Wang et al. (2022)'s approach is majority voting, where the defender splits the training data into multiple disjoint subsets, train models on them, and run majority voting of test inputs over those models. BagFlip Zhang et al. (2022) proposes a model-agnostic defense that uses the idea of randomized smoothing Cohen et al. (2019) studied in adversarial robustness. However, the certification offered by these works has only shown effective in toy datasets like MNIST, and it has been under-studied whether they will scale to practical scenarios, e.g., CIFAR10 or TinyImageNet. The computational demands of these defenses further hinder their deployment in practice.

## 3 DIFFUSION DENOISING AS CERTIFIED DEFENSE

We aim to certify the prediction of a model $f_\theta$ trained on a poisoned training data $D_{ptr}$. To formally define this goal, we define the perturbation bound $\epsilon$ as the $\ell_p$ distance between two datasets, computed by taking the sum across $\ell_p$ perturbations of all images. The perturbation space $P_\epsilon^\pi$ is the set of all the datasets $D_{ptr}$ obtained by tampering samples with the crafting algorithm $\pi$ with the bound $\epsilon$. Operating within the space $P$, for any test-time sample $(x_t, y_t)$, can produce a certificate that

$$\mathbf{Pr}_{D_{ptr} \in P_\epsilon^\pi(D_{tr})} \left[ f_{\theta \leftarrow D_{ptr}}(x_t) = f_{\theta \leftarrow D_{tr}}(x_t) \right] > 1 - \alpha$$

with arbitrarily high probability $1 - \alpha$, we guarantee that training on any poisoned dataset $D_{ptr} \in P_r^\pi(D_{tr})$ in the perturbation bound $\epsilon$ will classify the example $\mathbf{x}_t$ the same way as it was classified on the non-poisoned dataset $D_{tr}$.

### 3.1 ROBUSTNESS GUARANTEE

**Intuition.** Randomized smoothing Cohen et al. (2019) guarantees that a fixed classifier $f_\theta$ will classify any adversarial example $x'$ correctly with high probability. To make this argument, it first shows that even for extremely large values of noise $\delta$, the example $x + \delta$ will be classified correctly. Thus, the analysis of randomized smoothing is able to show that a (much smaller) worst-case direction $\delta_{adv}$ will not change the prediction of the classifier.

Here we make a similar argument, but change where the noise is added. Given a dataset $D_{ptr}$, where some examples may be slightly perturbed to poison the model, we want to guarantee that any model $f_\theta$ trained on $D_{ptr}$ will classify the test examples correctly. To achieve this, we train many models $f_\theta^i$ on datasets from $P_\delta^\pi$ (with extremely large perturbations $\delta$); if each of these classifiers consistently labels the test data correctly, then we can be guaranteed that (much smaller) worst-case poisoning directions $\delta_{adv}$ can not change the prediction of the classifier.

Randomized smoothing shows that adversarial noise to the test input $x$ will not cause misprediction because the predictions remain consistent even for large quantities of Gaussian noise $x + \delta$. In our

setting, adversarial noise to the model's training data $D_{tr}$ will not cause misprediction as even for large quantities of Gaussian noise $P_\delta^\pi$ to the training set the predictions of $f_{\theta \leftarrow D_{ptr}}$ remain consistent.

Randomized smoothing transforms a non-robust classifier $f$ into a smoothed (robust) classifier $g$:

$$g(x) = \arg \max_{c \in \mathcal{Y}} \mathbf{Pr}_{\delta \sim N(0, \sigma^2 I)}[f_\theta(x + \delta) = c].$$

But clean-label poisoning does not add any test-time perturbations to the target samples. Instead, it alters the training *dataset*, which then causes the *models* trained on the dataset to be different. Therefore, for our use-case we define:

$$g(x) = \arg \max_{c \in \mathcal{Y}} \mathbf{Pr}_{\theta \sim \text{train}(D_{ptr} \sim P_\delta^\pi)}[f_\theta(x) = c].$$

Each model $f_\theta$ is obtained by training a new classifier on a perturbed version of the training dataset.

## 3.2 (DIFFUSION) DENOISING FOR THE ROBUSTNESS

**Denoising diffusion probabilistic model (DDPMs)** are a recent generative model that works by learning the diffusion process of the form $x_t \sim \sqrt{1 - \beta_t} \cdot x_{t-1} + \beta_t \cdot \omega_t$, where $\omega_t$ is drawn from a standard normal distribution $\mathcal{N}(0, \mathbf{I})$ with $x_0$ sourced from the actual data distribution, and $\beta_t$ being fixed (or learned) variance parameters. This process transforms images from the target data distribution into purely random noise over time $t$, and the reverse *denoising* process constructs images in the data distribution, starting with random Gaussian noise. A DDPM with a fixed time-step $t \in \mathbb{N}^+$ and a fixed schedule samples a noisy version of a training image $x_t \in [-1, 1]^{w \cdot h \cdot c}$ of the form:

$$x_t := \sqrt{\alpha_t} \cdot x + \sqrt{1 - \alpha_t} \cdot \mathcal{N}(0, \mathbf{I}),$$

where $\alpha_t$ is a constant derived from $t$, which decides the level of noise to be added to the image (the noise increases consistently as $t$ grows). During training, the model minimizes the difference between $x$ and the denoised $x_t$, where $x_t$ is obtained by applying the noise at time-step $t$.

**Diffusion denoising for the robustness.** We utilize off-the-shelf DDPMs (Sohl-Dickstein et al., 2015; Ho et al., 2020; Nichol & Dhariwal, 2021) to *denoise* adversarial perturbations added to the training data and as a result, provide the robustness to clean-label poisoning. A naive adaptation of randomized smoothing to our scenario is to train multiple models on the training data with Gaussian noise augmentation or using adversarial training. But we need to add noise to the data that does not make it look natural or they are computationally demanding. We thus avoid using these approaches to train models, instead we want to remove the (potentially compromised) training set before training. Note that the robustness guarantee that randomized smoothing offers holds regardless of how a model is trained. We can still certify the robustness against clean-label poisoning. The fact that we use the denoising may make the guarantee a bit loose, but we show in our evaluation that the loose guarantee is sufficient to render existing attacks ineffective. Our denoising process is shown in Algorithm 1.

| **Pseudocode** Noise, denoise, train, classify | **Pseudocode** Randomized smoothing [Cohen et al.] |
|---|---|
| 1: **fn** DTCLASSIFY$(f, \sigma, D, x, n)$ | 1: **fn** PREDICT$(f, \sigma, D, x, n, \alpha)$ |
| 2: $\quad$ counts $\leftarrow \mathbf{0}$ | 2: $\quad$ counts $\leftarrow$ DTCLASSIFY$(f, \sigma, D, x, n_0)$ |
| 3: $\quad$ **for** $i \in \{1, 2, ..., n\}$ **do** | 3: $\quad \hat{c_A}, \hat{c_B} \leftarrow$ top two predictions in counts |
| 4: $\quad\quad t^*, \alpha_{t^*} \leftarrow$ GETTIMESTEP$(\sigma)$ | 4: $\quad n_A, n_b \leftarrow$ counts$[\hat{c_A}]$, counts$[\hat{c_B}]$ |
| 5: $\quad\quad \hat{D} \leftarrow$ NOISEANDDENOISE$(D, \alpha_{t^*}; t^*)$ | 5: $\quad$ **if** BINOMPVAL$(n_A, n_A + n_B, 0.5 \leq \alpha)$ **ret** $\hat{c_A}$ |
| 6: $\quad\quad \hat{f}_\theta \leftarrow$ TRAIN$(\hat{D}, f)$ | 6: $\quad$ **else ret** ABSTAIN |
| 7: $\quad\quad$ counts$[\hat{f}_\theta(x)] \leftarrow$ counts$[\hat{f}_\theta(x)] + 1$ | 7: |
| | 8: **fn** CERTIFY$(f, \sigma, D, x, n_0, n, \alpha)$ |
| 8: $\quad$ **ret** counts | 9: $\quad$ counts0 $\leftarrow$ DTCLASSIFY$(f, \sigma, D, x, n_0)$ |
| 9: | 10: $\quad \hat{c_A} \leftarrow$ top predictions in counts0 |
| 10: **fn** GETTIMESTEP$(\sigma)$ | 11: $\quad$ counts $\leftarrow$ DTCLASSIFY$(f, \sigma, D, x, n)$ |
| 11: $\quad t^* \leftarrow$ find $t$ s.t. $\frac{1 - \alpha_t}{\alpha_t} = \sigma^2$ | 12: $\quad p_A \leftarrow$ LOWERCFBOUND$($counts$[\hat{c_A}], $n$, 1 - \alpha)$ |
| 12: $\quad$ **ret** $t^*, \alpha_{t^*}$ | 13: $\quad$ **if** $p_A > 1/2$ **ret** $\hat{c_A}$ and radius $\sigma\Phi^{-1}(p_A)$ |
| | 14: $\quad$ **else ret** ABSTAIN |

### 3.3 PREDICTION AND CERTIFICATION

Now we present algorithms for running a prediction with the classifiers trained on $D_{ptr}$ and certifying the robustness of the model's prediction on a test-time sample $x$. Our work extends the algorithms presented by Cohen et al. (2019) to clean-label poisoning settings, as shown in Algorithm 2.

**Prediction.** To compute the prediction for a test-time sample $x$, we make an adaptation of the standard randomized smoothing. We train $n$ classifiers on $n$ different noised-then-denoised training datasets, and return the predictions of these classifiers on the target $x$, with the algorithm DTCLASSIFY(·). We run this algorithm for a sufficient number of times (e.g., over 1000 times). The prediction output is then the majority-voted label from these classifiers.

**Certification** process exactly follows the standard randomized smoothing. We first count the number of occurrences of the most likely label $\hat{c_A}$ compared to any other runner-up label, and from this can derive a radius (for the training set perturbations) on which $x$ is guaranteed to be robust.

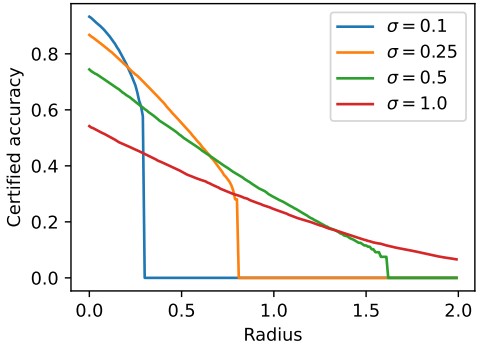
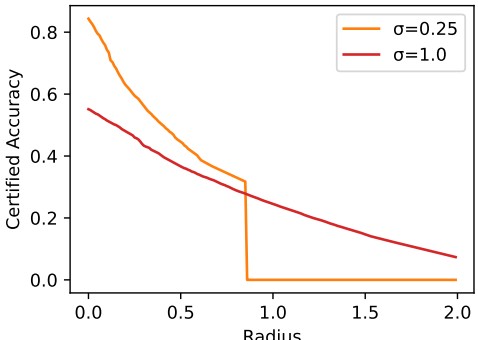

Figure 1: **Certified radius and accuracy** attained by *denoising* the CIFAR10 training data with different $\sigma$ values in {0.1, 0.25, 0.5, 1.0}.

Figure 2: **Certified radius and accuracy** attained by adding *Gaussian random noise* to training data with different $\sigma$ in {0.1, 0.25, 0.5, 1.0}.

**Results.** We train 10k CIFAR10 models using an efficient training pipeline that trains a single model in 15 seconds on an A100 GPU (for a total of 1.7 GPU days of compute). These models have a clean accuracy of between $92\%$ and $58\%$ depending on the level of noise $\sigma$ introduced, and for the first time can certify non-trivial robustness to clean-label poisoning attacks. A full curve comparing certified accuracy for all given perturbation radii is shown in Figure 1. Instead of applying a diffusion model, we test the randomized smoothing approach proposed by Weber et al. (2023): adding Gaussian random noise directly to the images and train on these (noisy) images. This achieves a similar level of certified accuracy and robustness as our denoising approach, but at a cost of training for $20\times$ longer.

### 3.4 COMPUTATIONAL EFFICIENCY FOR CERTIFICATION

In order to obtain a robustness certificate, we must train many CIFAR10 models on different denoisings of the training dataset. This step is computationally expensive, but is in line with the computational complexity of prior randomized smoothing based approaches that require similar work. However, as we will show later, in practice we can empirically obtain robustness with only a very small number of training runs (even as low as **one**!), confirming the related observation from (Lucas et al., 2023) who find a similar effect holds for randomized smoothing of classifiers.

## 4 EMPIRICAL EVALUATION

We empirically evaluate the effectiveness of our certified defense against clean-label poisoning. We adapt the poisoning benchmark developed by Schwarzschild et al. (2021). This benchmark runs clean-label poisoning attacks with the same attack configurations, allowing us to compare our defense's effectiveness across many existing attacks.

**Poisoning attacks.** We evaluate our defense against seven clean-label poisoning attacks: four targeted poisoning (Poison Frogs (Shafahi et al., 2018), Convex Polytope (Zhu et al., 2019), Bullseye Polytope (Aghakhani et al., 2021), Witches' Brew (Geiping et al., 2021b)) and three backdoor attacks

(Label-consistent Backdoor (Turner et al., 2019), Hidden-Trigger Backdoor (Saha et al., 2020), and Sleeper Agent (Souri et al., 2022)). The first three attacks operate in the transfer-learning scenarios (i.e., they assume that the victim fine-tunes a model from some given initialization), whereas the rest assumes the victim trains a model from-scratch. While our defense offers certificates only to targeted poisoning attacks, we test if the defense can mitigate clean-label backdooring like the prior work did.

**Metrics.** We employ two metrics: clean accuracy and attack success rate. We compute the accuracy on the entire test set. An attack is successful if a poisoned model classifies a target sample to the intended class. When evaluating backdoor attacks, we only count the cases where a tampered model misclassifies the target sample containing a trigger pattern.

**Methodology.** We run our experiments in CIFAR10 and Tiny ImageNet (Krizhevsky et al., 2009; Le & Yang, 2015). Following the standard practices in the prior work (Schwarzschild et al., 2021), we run each attack 100 times on different targets and report the averaged metrics over the 100 runs. We randomly choose the base and target classes. We set the poisoning budget to 0.5–1% for all our attacks: the attacker can tamper with only 500 samples of the entire 50k training instances in CIFAR10 and 250 of the total 100k training samples in Tiny ImageNet.

**Attack scenarios.** We evaluate against both the transfer-learning and training from-scratch attacks. In transfer-learning, we fine-tune a pre-trained model on the tampered training set while we train a randomly initialized model in the training from-scratch scenario. We also evaluate against the attackers with two types of knowledge: white-box and black-box. The white-box attackers know the victim model's internals, while the black-box attackers do not. More details are in Appendix A.

## 4.1 Effectiveness of Our Denoising Defense

We first show to what extent our defense makes the seven poisoning attacks ineffective. Our defense is attack-agnostic. A defender does not know whether the training data contains poisoning samples or not; instead, the defender always trains a model on the denoised training set. We measure the trained model's accuracy and attack success and compare them with the baseline, where a model is trained directly without denoising. We evaluate with different defense strengths: $\sigma \in \{0.1, 0.25, 0.5, 1.0\}$. The attacks are with the perturbation bound of 16-pixels in $\ell_\infty$-norm and that of 3.5–6.5 in $\ell_2$-norm. We extended many poisoning attacks for supporting $\ell_2$ bounds. Both $\ell_\infty$ and $\ell_2$ bounds we use are comparable to each other. We show the conversion between the two bounds in Appendix B. Due to the page limit, the results from Tiny ImageNet and attacking with different $\ell_p$-norms are in Appendix C.

Table 1: **Defense effectiveness in transfer-learning scenarios (CIFAR10).** We measure the clean accuracy and attack success of the models trained on the denoised training set. Each cell shows the accuracy in the parentheses and the attack success outside. Note that [†] indicates the runs with $\sigma$=0.0, the same as our baseline that trains models without any denoising.

| Poisoning attacks | Knowledge | Our defense against $\ell_2$ attacks at $\sigma$ (%) | | | | | Our defense against $\ell_\infty$ attacks at $\sigma$ (%) | | | | |
|---|---|---|---|---|---|---|---|---|---|---|---|
| | | [†]0.0 | 0.1 | 0.25 | 0.5 | 1.0 | [†]0.0 | 0.1 | 0.25 | 0.5 | 1.0 |
| Poison Frog! | WB | (93.6)99.0 | (93.3)0.0 | (91.8)1.0 | (84.8)0.0 | (79.9)1.0 | (93.6)68.8 | (93.3)0.0 | (92.7)0.0 | (90.8)0.0 | (87.4)0.0 |
| Convex Polytope | | (93.7)16.2 | (93.2)0.0 | (91.7)0.0 | (86.6)0.0 | (77.0)0.0 | (93.7)12.2 | (93.3)0.0 | (92.7)0.0 | (90.8)1.0 | (87.5)0.0 |
| Bullseye Polytope | | (93.5)100 | (93.3)4.0 | (92.6)0.0 | (87.5)0.0 | (79.2)1.0 | (93.5)100 | (93.3)0.0 | (92.7)0.0 | (90.8)1.0 | (87.5)0.0 |
| Label-consistent Backdoor | | - | | | | | (93.2)1.0 | (93.3)0.0 | (92.6)0.0 | (90.8)1.0 | (87.5)0.0 |
| Hidden Trigger Backdoor | | - | | | | | (93.4)7.0 | (93.3)0.0 | (92.6)0.0 | (90.8)0.0 | (87.5)0.0 |
| Poison Frog! | BB | (91.6)10.0 | (91.2)0.0 | (89.6)0.5 | (82.9)0.0 | (77.8)2.0 | (91.7)2.5 | (91.3)0.0 | (90.3)0.0 | (88.8)0.5 | (86.2)1.0 |
| Convex Polytope | | (91.7)3.0 | (91.0)0.0 | (89.5)0.0 | (84.6)0.5 | (73.6)1.0 | (91.8)2.5 | (91.3)0.0 | (90.3)0.0 | (88.8)0.5 | (86.2)1.0 |
| Bullseye Polytope | | (91.6)9.0 | (91.3)0.0 | (90.3)0.0 | (85.5)0.0 | (76.3)1.0 | (91.6)8.0 | (91.3)0.0 | (90.3)0.0 | (88.8)0.5 | (86.2)0.5 |
| Label-consistent Backdoor | | - | | | | | (91.5)1.0 | (91.3)0.0 | (90.3)0.0 | (88.8)0.0 | (86.2)1.5 |
| Hidden Trigger Backdoor | | - | | | | | (91.6)4.0 | (91.2)1.0 | (90.3)1.0 | (89.3)1.5 | (86.3)1.5 |

**Transfer-learning scenarios.** Table 1 summarizes our results. Our defense is also effective in mitigating the attacks in transfer-learning. Each cell contains the accuracy and the attack success rate of a model trained on the denoised training set. All the numbers are the averaged values over 100 attacks. Against the white-box attacks, we reduce the success rate of the targeted poisoning attacks to 0–4% at $\sigma$ of 0.1. In the black-box attacks, the targeted poisoning is less effective (with 3–10% success), but still, the defense reduces the success to 0–1%. We observe that the two clean-label backdoor attacks show 1–7% success regardless of the attacker's knowledge. We thus exclude these attacks from the subsequent evaluation. Using $\sigma$ greater than 0.5 increases the attack success by 0.5–2%. But, this may not because of the successful poisoning attacks but due to the significant

decrease in a model's accuracy (3-9%). Increasing $\sigma$ may defeat a stronger attack but significantly decrease a model's accuracy. This result is interesting, as in our theoretical analysis shown in Sec 3, our defense needs $\sigma$ of 0.25, while 0.1 is sufficient to defeat both the attacks.

**Training from-scratch scenarios.** Table 2 shows our results with the same format as in Table 1. We demonstrate that our defense mitigates both attacks at $\sigma = 0.25$, aligned to what we've seen in our theoretical analysis. In Witches' Brew, the attack success decreases from 34–71% to 3–10%; against Sleeper Agent, the attack success decreases from 19–40% to 7–19%[1]. Note that to reduce their attack success to ~10%, our defense needs a minimum $\sigma$ of 0.25–0.5. It is also noteworthy that our defense reduces the backdooring success significantly while the certificate does not hold because test-time samples are altered with trigger patterns.

Table 2: **Defense effectiveness in training from-scratch scenarios (CIFAR10).** We measure the accuracy and attack success of the models trained on the denoised training set. Each cell shows the accuracy in the parentheses and the attack success outside. Note that $^\dagger$ indicates the runs with $\sigma$=0.0, the same as our baseline that trains models without any denoising. We use an ensemble of four models, and WB and BB stand for the white-box and the black-box attacks, respectively.

| Poisoning attacks | Knowledge | Our defense against $\ell_2$ attacks at $\sigma$ (%) | | | | | Our defense against $\ell_\infty$ attacks at $\sigma$ (%) | | | | |
|---|---|---|---|---|---|---|---|---|---|---|---|
| | | $^\dagger$0.0 | 0.1 | 0.25 | 0.5 | 1.0 | $^\dagger$0.0 | 0.1 | 0.25 | 0.5 | 1.0 |
| Witches' Brew | WB | (92.2)71.0 | (86.4)54.0 | (72.3)10.0 | (46.7)11.0 | (42.5)10.0 | (92.3)65.0 | (86.5)9.0 | (71.9)3.0 | (46.0)9.0 | (41.3)7.0 |
| Sleeper Agent Backdoor | | (92.4)40.5 | (84.4)66.5 | (71.8)19.5 | (46.8)13.0 | (39.9)10.5 | (92.4)35.0 | (86.1)17.0 | (73.0)8.0 | (47.0)9.5 | (39.7)10.0 |
| Witches' Brew | BB | (90.1)45.5 | (85.9)28.0 | (75.5)4.0 | (58.8)7.0 | (49.0)10.0 | (90.0)33.5 | (85.8)3.5 | (75.5)2.5 | (58.8)6.0 | (48.7)6.5 |
| Sleeper Agent Backdoor | | (90.0)39.5 | (85.0)44.5 | (75.1)14.5 | (58.6)9.5 | (49.0)8.0 | (90.0)18.5 | (85.6)11.5 | (75.5)7.0 | (58.8)8.0 | (48.4)8.0 |

## 4.2 Improving Model Utility

A shortcoming of certified defenses (Levine & Feizi, 2020; Wang et al., 2022; Zhang et al., 2022) is that the certified accuracy is substantially lower than the undefended model's accuracy. We also observe in §4.1 that our defense, when we train a model from scratch on the data, denoised with large $\sigma$ values, is not free from the same issue, which hinders their deployment in practice.

In §3, we show that our defense is agnostic to how we train a model, including how we *initialize* a model's parameters. We leverage this property and minimize the utility loss by initializing model parameters in specific ways before we train the model on the denoised data. We test two scenarios: a defender can use in-domain data or out-of-domain data to pre-train a model and use its parameters to initialize. We evaluate them against the three most effective attacks shown in Table 1.

Table 3: **Improving the utility of defended models.** We show the accuracy of defended models and the attack success after employing our two strategies. The top three rows are the baselines from Table 1 and 2, and the next two sets of three rows are our results. We highlight the cells showing the accuracy improvements in **bold**. ResNet18 models are used. BP, WB, and SA indicate Bullseye Polytope, Witches' Brew, and Sleeper Agent, respectively.

| Att. | Initialization | Our defense against $\ell_\infty$ attacks at $\sigma$ (%) | | | | |
|---|---|---|---|---|---|---|
| | | $^\dagger$0.0 | 0.1 | 0.25 | 0.5 | 1.0 |
| BP | N/A | (93.5)100 | (93.3)0.0 | (92.7)0.0 | (90.8)1.0 | (87.6)0.0 |
| WB | (Baseline) | (92.3)65.0 | (86.5)9.0 | (71.9)3.0 | (46.0)9.0 | (41.3)7.0 |
| SA | | (92.4)34.0 | (86.2)18.5 | (73.0)7.9 | (47.2)8.2 | (40.1)11.5 |
| BP | In-domain | (93.5)100 | (93.3)0.0 | (92.6)1.0 | (**90.9**)1.0 | (**87.7**)2.0 |
| WB | (CIFAR10) | (92.3)65.0 | (86.5)9.0 | (**72.7**)5.0 | (**47.2**)7.0 | (**38.6**)8.0 |
| SA | | (92.4)34.0 | (86.2)18.1 | (**73.4**)8.5 | (**47.8**)9.9 | (**36.9**)11.2 |
| BP | $^\dagger$Out-of-domain | (85.2)4.0 | (71.1)7.0 | (66.4)5.0 | (58.6)5.0 | (48.7)6.0 |
| WB | (ImageNet-21k) | (86.6)14.0 | (84.1)4.0 | (**79.0**)0.0 | (**67.7**)5.0 | (**54.0**)6.0 |
| SA | | (92.3)35.0 | (86.1)18.0 | (72.9)8.5 | (47.6)10.0 | (37.4)10.0 |

$^\dagger$ResNet18 in Torchvision library; only the latent space dimension differs.

**Initializing a model using in-domain data.** Our strategy here is an adaptation of warm-starting (Ash & Adams, 2020). We first train a model from scratch on the tampered training data for a few epochs to achieve high accuracy. It only needs 5–10 epochs in CIFAR10. We then apply our defense and continue training the model on the denoised training set.

Table 3 summarizes our results. The middle three rows are the results of leveraging the in-domain data. We train the ResNet18 models on the tampered training set for 10 epochs and continue training on the denoised training data for the rest 30 epochs. Our first strategy (warm-starting) increases the accuracy of the defended models while defeating clean-label poisoning attacks. Under strong defense guarantees $\sigma > 0.1$, the models have 0.5–2.2% increased

---

[1]The success rate of typical backdoor attacks is ~90%; thus, the success below 19% means they are ineffective.

accuracy, compared to the baseline, while keeping the attack success ∼10%. In $\sigma = 0.1$, we achieve the same accuracy and defense successes. Our strategy can be potentially useful when a defender needs a stronger guarantee, such as against stronger clean-label poisoning future work will develop.

**Using models pre-trained on out-of-domain data.** Now, instead of running warm-starting on our end, we can also leverage "warm" models available from the legitimate repositories. To evaluate, we take the ResNet18 model pre-trained on ImageNet-21k. We use this pre-trained model in two practical ways. We take the model as-is and train it on the denoised training data. In the second approach, we combine the first approach with the previous idea. We train the model on the tampered training data for a few epochs and then train this fine-tuned model on the denoised data.

The bottom three rows of Table 3 are the results of using the warm model. We fine-tune the ResNet18 for 40 epochs on the denoised training data. In many cases, our second strategy can improve the accuracy of the defended models trained with strong defense guarantees ($\sigma > 0.1$). These models achieve 6.5–13.2% greater accuracy than the results shown in Table 2 and 1. In $\sigma = 1.0$, the final accuracy of defended models has a negligible difference from the baseline. We are the first work to offer practical strategies to manage the utility-security trade-off in certified defense.

### 4.3 COMPARISON TO EXISTING POISONING DEFENSES

We finally examine how our defense works better/worse than existing defenses. We compare ours with five defenses: k-NN (Peri et al., 2020), DP-SGD (Ma et al., 2019; Hong et al., 2020), AT (Geiping et al., 2021a), FrieNDs (Liu et al., 2022), and ROE (Rezaei et al., 2023). The ROE is a certified defense, and the other four are non-certified ones. We use $\ell_\infty$-norm of 16 in CIFAR10.

Table 4: **Comparison of ours to Deep kNN.**

| Poisoning attacks | TP | FP | Trained | Ours ($\sigma = 0.1$) |
|---|---|---|---|---|
| Witches' Brew | 31 | 464 | (86.3) 3.0 | (86.5) 9.0 |
| Bullseye Polytope | 431 | 63 | (93.7) 25.0 | (93.3) 0.0 |

**Deep kNN** removes poisons from the training data by leveraging the k-nearest-neighbor (kNN) algorithm. They run kNN on the latent representations of the training data to identify potentially malicious samples (i.e., samples with labels different from the nearest ones) and remove them.

We compare Deep kNN's effectiveness to our defense. We use the defense configurations that bring the best results in the original study, i.e., setting $k$ to 500. Table 5 shows our results. Deep kNN fails to remove most poisoning samples (but it removes many benign samples!).

**DP-SGD.** Ma et al. (2019) proposed a certified defense against poisoning attacks that leverage differential privacy (DP). DP makes a model less sensitive to a single-sample modification to the training data. But the defense offers a weak guarantee in practice; for example, the certificate is only valid in CIFAR10, when an adversary tampers one training sample. (Hong et al., 2020) later empirically shows that DP is still somewhat effective against poisoning. We compare our defense to the training with $(1.0, 0.05)$-DP, follow the prior work (Lecuyer et al., 2019). We use Opacus[2] to train models with DP and keep all the other configurations the same. In Table 5, both defenses significantly reduce the attack success to 0–16%. Our defense achieves at most 10% higher accuracy than DP. DP reduces the attack success slightly more against clean-label backdooring.

Table 5: **Comparison of ours to training-time defenses.**

| Poisoning attacks | Heuristic defense | | | Certified defense | |
|---|---|---|---|---|---|
| | DP-SGD | AT | FrieNDs | ROE | Ours |
| Bullseye Polytope | (93.9) 7.0 | (90.3) 96.0 | (90.2) 10.0 | *N/A | (93.5) 0.0 |
| Witches' Brew | (76.0) 4.0 | (66.5) 2.0 | (87.6) 8.0 | (70.2) 10.0 | (86.5) 9.0 |
| Sleeper Agent | (74.8) 5.0 | (69.0) 6.0 | (87.4) 14.0 | (68.6) 12.0 | (86.0) 17.0 |

*ROE is incompatible with BP as ROE needs to train 250 models *from scratch*.

**AT.** Geiping et al. (2021a) adapts the adversarial training (AT) for defeating clean-label poisoning: in each mini-batch, instead of crafting adversarial examples, they synthesize poisoning samples and have a model to make correct classifications on them. While effective, the model could overfit a specific poisoning attack used in the adapted AT, leaving the model vulnerable to unseen poisoning attacks. We thus compare our defense with the original AT with the PGD-7 bounded to $\epsilon$ of 4, assuming that clean-label poisons are already adversarial examples. AT thus may not add more perturbations to them during training. Table 5 shows that our defense achieves higher accuracy than the robust models while making the attcaks ineffective. Interestingly, AT cannot defeat the BP attacks.

---

[2]https://opacus.ai

**FrieNDs.** Liu et al. (2022) advances the idea of training robust models. Instead of adding random Gaussian noise to the data during training, they use "friendly" noise, pre-computed by a defender, that minimizes the accuracy loss to a model they will train. When their defense is in action, they train a model on the tampered training data with two types of noise: the friendly noise computed before and a weak random noise sampled uniformly within a bound. Ours and FrieNDs greatly reduce the attack success while preserving the model's utility. The critical difference is that ours is a certified defense, while FrieNDs is not. A future work we leave is developing adaptive attacks against FrieNDs.

**ROE.** Certified defenses (Steinhardt et al., 2017; Ma et al., 2019; Diakonikolas et al., 2019; Levine & Feizi, 2020; Gao et al., 2021; Zhang et al., 2022; Wang et al., 2022; Rezaei et al., 2023) are; however, most of these defenses showed their effectiveness in limited scenarios, such as in MINST-17 (Zhang et al., 2022), a subset of MNIST containing only the samples of the digits 1 and 7. In consequence, they aren't compatible with our clean-label poisoning settings. A recent defense by Rezaei et al. (2023) demonstrates the certified robustness in practical settings; we thus compare our defense to their run-off-election (ROE). In Table 5, we empirically show that our defense is comparable to ROE while we preserve the model accuracy more.

**Note on the computational efficiency.** Most training-time defenses require additional mechanisms we need to "add" to the training algorithm. AT needs to craft adversarial examples or clean-label poisons (Geiping et al., 2021a). DP-SGD introduces noise into the gradients during training. FrieNDs pre-computes friendly noise and then optimally applies the noise during the training, and ROE requires training of 250 base classifiers to achieve the improved certificates. In contrast, our defense only requires to run forward *once* with an off-the-shelf diffusion model.

## 5 DISCUSSION

Our work offers a new perspective on the cat-and-mouse game played in clean-label poisoning.

**Initial claims from the cat:** Clean label poisoning attacks, with human-imperceptible perturbations that keeping the original labels, are an attractive option to inject malicious behaviors into models. Even if a victim carefully curates the training data, they remain effective by evading *filtering* defenses that rely on statistical signatures distinguishing clean samples from poisons (Nelson et al., 2008a; Suciu et al., 2018; Peri et al., 2020).

**The mouse's counterarguments:** Yet, as seen in research on the adversarial robustness, such small perturbations can be *brittle* (Carlini et al., 2022): by adding a large quantity of noise and then denoising the images, the adversarial perturbations become nearly removed—and so clean-label perturbations are also removed. In prior work (and in this work as well), we leverage this observation and propose both certified and non-certified heuristic defenses.

**Additional arguments from the cat:** A counterargument from the clean-label attack's side was that those defenses inevitably compromise a model's performance—a fact corroborated by the prior work on the adversarial robustness (Tsipras et al., 2018; Zhang et al., 2019). Similarly, defenses could greatly reduce the poisoning attack success, but at the same time, they decrease the accuracy of defended models, often tipping the scales in favor of the adversary. If a defense yields a CIFAR-10 model with 60–70% utility, what is the point of having such models in practice? Indeed, our own certified models require degraded utility to achieve certified predictions. Other defenses, e.g., DP-SGD (Ma et al., 2019; Hong et al., 2020), are computationally demanding, increasing the training time an order of magnitude.

Our work shows that this cat-and-mouse game need not be as pessimistic as the cat portrays.

By leveraging an off-the-shelf DDPMs (Ho et al., 2020; Nichol & Dhariwal, 2021), we can purify the training data, possibly containing malicious samples, *offline* and render six clean-label attacks ineffective. With a weak provable guarantee against the training data contamination ($\ell_\infty$-norm of 2), the majority of attacks reach to 0% success. A few recent attacks exhibit 0–10% success rate—a rate comparable to random misclassification of test-time inputs.

In contrast to the cat's counterargument, we can also address the accuracy degradation problem—as long as we do not need certified robustness. Existing defenses that work to defend against poisoning require applying a particular training algorithm designed by the defender (Hong et al., 2020; Geiping et al., 2021a; Liu et al., 2022; Levine & Feizi, 2020; Wang et al., 2022; Zhang et al., 2022). By

decoupling the training process and the defense, we open a new opportunity to develop training techniques that can enhance the accuracy of the defended models while keeping the robustness. Our results propose two effective heuristic algorithms in improving defended models' utility, especially when the models are under strong defense guarantees (i.e., $||\epsilon||_\infty > 2$).

**Reproducibility Statement.** To ensure our work is reproducible, we provide descriptions of the datasets, models, training hyper-parameters, poisoning attacks and defenses both in the main text and Appendix. Specifically, Sec 4 offers detailed discussion on the poisoning attacks, attack scenarios evaluation metrics, and methodology. Appendix A provides the implementation details, training hyperparameters, and poisoning attack configurations. We believe this detailed information will facilitate the successful replication of our work. We also include the source code in our submission.

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

## A  EXPERIMENTAL SETUP IN DETAIL

We adapt the poisoning benchmark from the prior work (Schwarzschild et al., 2021). Most work on clean-label poisoning attacks uses this benchmark for showcasing their attack success. The open-sourced implementation of Sleeper Agent is incompatible with this framework; thus, we run this attack separately. We implement the attacks to use a comparable $\ell_2$-norm bound in the poison-crafting process (see §B). We implemented our benchmarking framework in Python v3.7[3] and PyTorch v1.13[4]. We use the exact attack configurations and training hyper-parameters that the original study employs.

In contrast to the original work, we made two major differences. We first increase the perturbations bounded to $||\epsilon||_\infty = 16$ as the 8-pixel bound attacks do not result in a high attack success rate. Defeating 8-pixel bounded attacks is trivial for any poisoning defenses. Second, we do not use their fine-tuning subset, which only contains 2500 training samples and 25–50 poisons. It (as shown in the next subsections) leads to 60–70% accuracy on the clean CIFAR10 test-set, significantly lower than 80–90%, which can be trivially obtained with any models and training configurations. If a model trained on the contaminated training data misclassifies a target, it could be a mistake caused by a poorly performing model.

We run a comprehensive evaluation. We run 7 attacks; for each attack, we run 100 times of crafting and training/fine-tuning a model. We also examine 5 different denoising factor $\sigma$. In total, we ran 3500 poisoning attacks. For the most successful three attacks, we run 6 different certified and non-certified defenses over 100 poisoning runs. The three training-time defenses (DP-SGD, AT, FrieNDs) require 100 trainings of a model for each, and the certified defense (RoE) requires $100 \times 250$ trainings in total. To accommodate this computational overhead, we use two machines, each equipped with 8 Nvidia GPUs. Crafting a poisoning set takes approximately one hour and training takes 30 minutes on a single GPU.

## B  TRANSLATING $l_\infty$-BOUND INTO $l_2$-BOUND

The certification we offer in Sec 3 is defined in $l_2$-norm, while most poisoning attacks work with $l_\infty$-norm, e.g., of 8 or 16. We therefore convert these $l_\infty$-bounds into $l_2$-bounds. We assume the worst-case perturbation in the $l_\infty$-space that changes every pixel location of an image by 16 pixels, compute the $l_2$-norm of that perturbation as follows, and use it in §4:

- $l_\infty$-bound of 8 pixels: $\sqrt{3 * 32 * 32 * (8/255)^2} = 1.74$
- $l_\infty$-bound of 16 pixels: $\sqrt{3 * 32 * 32 * (16/255)^2} = 3.48$

Oftentimes, the $\ell_2$ attacks with the comparable bound 3.48 do not lead to comparable attack success. We, therefore, especially for the poisoning attacks in the transfer-learning scenarios, increase the bound to 6.43. We note that most existing attacks either do not implement $\ell_2$ bounds or are unbounded (Shafahi et al., 2018).

## C  MORE EXPERIMENTAL RESULTS

Here we include additional experimental results.

### C.1  USING THE EVALUATION SETUP BY SCHWARZSCHILD ET AL. (2021)

We examine the weaker adversary whose perturbation is bounded to $||\delta||_\infty = 8$. We also employ the exact setup as the prior work (Schwarzschild et al., 2021), where we craft 25 poisons on a ResNet18 pre-trained on CIFAR-100 and fine-tune the model on a subset of the CIFAR-10 training data. The subset contains the first 250 images per class (2.5k samples in total).

Table 6 summarizes our results against clean-label poisoning attacks with $l_\infty$-norm of 8. We examine five poisoning attacks in the white-box and black-box scenarios. We focus on the attacks against transfer-learning as the specific data splits the prior work (Schwarzschild et al., 2021) uses are

---

[3]https://www.python.org/
[4]https://pytorch.org

Table 6: **Diffusion denoising against clean-label poisoning (CIFAR10).** We denoise the $l_\infty$-norm of 8 perturbations added by five poisoning attacks by running a single-step stable diffusion on the entire training set. In each cell, we show the average attack success over 100 runs and the average accuracy of models trained on the denoised data in the parenthesis. Note that $^\dagger$ indicates the runs with $\sigma$=0.0, the same as our baseline that trains models without any denoising.

| Poisoning attacks | Scenario | Our defense against $\ell_\infty$ attacks at $\sigma$ (%) | | | | |
|---|---|---|---|---|---|---|
| | | $^\dagger$0.0 | 0.5 | 1.0 | 1.5 | 2.0 |
| Poison Frog! (Shafahi et al., 2018) | White-box | $^{(69.8)}$13.0 | $^{(55.9)}$8.0 | $^{(43.1)}$4.0 | $^{(34.7)}$9.0 | $^{(26.9)}$11.0 |
| Convex Polytope (Zhu et al., 2019) | | $^{(69.8)}$24.0 | $^{(53.8)}$5.0 | $^{(42.6)}$9.0 | $^{(35.0)}$8.0 | $^{(27.7)}$6.0 |
| Bullseye Polytope (Aghakhani et al., 2021) | | $^{(69.4)}$100.0 | $^{(55.8)}$10.0 | $^{(42.8)}$8.0 | $^{(35.0)}$9.0 | $^{(27.4)}$9.0 |
| Label-consistent Backdoor (Turner et al., 2019) | | $^{(69.8)}$2.0 | $^{(55.9)}$3.0 | $^{(42.7)}$5.0 | $^{(34.9)}$8.0 | $^{(27.4)}$12.0 |
| Hidden Trigger Backdoor (Saha et al., 2020) | | $^{(69.8)}$5.0 | $^{(55.9)}$3.0 | $^{(42.7)}$10.0 | $^{(35.2)}$9.0 | $^{(27.3)}$9.0 |
| Poison Frog! (Shafahi et al., 2018) | Black-box | $^{(67.9)}$7.0 | $^{(53.8)}$6.0 | $^{(43.3)}$2.5 | $^{(35.3)}$8.0 | $^{(28.8)}$6.5 |
| Convex Polytope (Zhu et al., 2019) | | $^{(67.9)}$4.0 | $^{(53.7)}$3.0 | $^{(43.3)}$3.5 | $^{(35.2)}$3.0 | $^{(28.8)}$8.0 |
| Bullseye Polytope (Aghakhani et al., 2021) | | $^{(67.7)}$8.0 | $^{(53.8)}$17.5 | $^{(43.2)}$16.5 | $^{(35.3)}$7.5 | $^{(28.6)}$8.5 |
| Label-consistent Backdoor (Turner et al., 2019) | | $^{(67.9)}$3.5 | $^{(53.8)}$2.0 | $^{(43.5)}$4.5 | $^{(35.2)}$2.5 | $^{(29.0)}$8.0 |
| Hidden Trigger Backdoor (Saha et al., 2020) | | $^{(67.9)}$7.5 | $^{(53.7)}$2.0 | $^{(43.3)}$7.0 | $^{(35.2)}$10.5 | $^{(28.7)}$8.5 |

only compatible with them. Poisoning attacks against training from scratch scenarios use the entire CIFAR10 as ours, so we don't need to duplicate the experiments. In the white-box setting, we fine-tune the CIFAR-100 ResNet18 model (that we use for crafting poisons) on the poisoned training set. In the black-box setting, we fine-tune different models (VGG16 and MobileNetV2 models pre-trained on CIFAR-100) on the same poisoned training set. We use $\sigma$ in $\{0.5, 1.0, 1.5, 2.0\}$ for our single-step DDPM. $\sigma$=0.0 is the same as no defense.

**Results.** Diffusion denoising significantly reduces the poisoning attack success. The most successful attack, Bullseye Polytope in the white-box setting, achieves the attack success of 100% in $l_\infty$-norm of 8 pixels, but denoising with $\sigma$ of 0.5 can reduce their success to 10%. Our defense reduces the attack success of Poison Frog! and Convex Polytope from 13-34% to 2-8% at $\sigma = 0.5$. The two backdoor attacks (label-consistent and hidden trigger) exploiting clean-label poisoning are not successful in the benchmark setup (their success rate ranges from 2-7.5%). We thus could not quantify our defense's effectiveness against these backdoor attacks. Note that Schwarzschild et al. (2021) also showed these backdoor attacks ineffective, and our finding is consistent with their results. We exclude them from our Tiny ImageNet results in §C.2.

We also observe that the increased $\sigma$ (strong denoising) can significantly reduce the utility of a model trained on the denoised training data. Note that since the setup uses only 2.5k CIFAR10 samples for fine-tuning, and in consequence, the model's utility is already ~70% at most, much lower than our setup. As we increase the $\sigma$ from 0.5 to 2.0, the fine-tuned model's accuracy leads to 56% to 27%. However, we show that with the small $\sigma$, our diffusion denoising can reduce the attack success significantly. We also show in our evaluation (§4) that we achieve a high model's utility while keeping the same $\sigma$. We attribute the increased utility to recent model architectures, such as VisionTransformers, or to pre-training a model on a larger data corpus. We leave further investigation for future work.

Moreover, in a few cases, the poisoning success increases from 2–7% to 10–13% as we increase $\sigma$. We attribute this increase not to the attack being successful with a high $\sigma$ but to the poor performance of a model. For example, the accuracy of a model with $\sigma = 2.0$ is ~27%, meaning that four out of five targets in a class can be misclassified.

## C.2 TINY IMAGENET RESULTS

We also ran our experiments with Tiny ImageNet to examine whether our findings are consistent across different datasets. We assume the same adversary who can add perturbations bounded to $\ell_\infty$-norm of 16 pixels. For the three attacks in transfer-learning scenarios, we craft 250 poisoning samples on a VGG16 pret-trained on the same dataset and fine-tune the model on the tampered training set. We do not evaluate backdoor attacks because they are either ineffective (label-consistent and hidden trigger) or the original study did not employ Tiny ImageNet (Sleeper Agent).

Table 7: **Diffusion denoising against clean-label poisoning (Tiny ImageNet).** We consider four attacks with the $\ell_\infty$-norm of 16 perturbation bound. In each cell, the attack success and accuracy of models trained on the denoised data in the parenthesis on average over 100 runs. Note that $\dagger$ indicates the runs with $\sigma$=0.0, representing no-defense scenario.

| | | Our defense against $\ell_\infty$ attacks at $\sigma$ (%) | | | | |
|---|---|---|---|---|---|---|
| Poisoning attacks | Scenario | $^\dagger$0.0 | 0.1 | 0.25 | 0.5 | 1.0 |
| Poison Frog! (Shafahi et al., 2018) | WB | $^{(58.9)}$79.0 | $^{(56.3)}$3.0 | $^{(53.3)}$2.0 | $^{(46.5)}$2.0 | $^{(32.0)}$1.0 |
| Convex Polytope (Zhu et al., 2019) | | $^{(58.9)}$95.0 | $^{(56.3)}$14.6 | - | - | - |
| Bullseye Polytope (Aghakhani et al., 2021) | | $^{(58.8)}$100.0 | $^{(57.4)}$100.0 | - | $^{(48.7)}$0.0 | - |
| Poison Frog! (Shafahi et al., 2018) | BB | $^{(58.0)}$5.0 | $^{(51.9)}$0.0 | $^{(46.5)}$0.0 | $^{(36.0)}$1.0 | $^{(22.6)}$1.0 |
| Convex Polytope (Zhu et al., 2019) | | $^{(58.1)}$0.0 | $^{(51.9)}$0.0 | - | - | - |
| Bullseye Polytope (Aghakhani et al., 2021) | | $^{(57.9)}$9.0 | $^{(59.1)}$33.3 | - | $^{(49.8)}$16.7 | - |

**Results.** Table 7 shows our results. Our results are consistent with what we observe in CIFAR10. Note that the poisoning attacks are more successful against Tiny ImageNet, and we attribute the success to the learning complexity of Tiny ImageNet over CIFAR10: Tiny ImageNet has 200 classes and 100k training samples. Prior work (Schwarzschild et al., 2021) had the same observation. Our defense significantly reduces the attack success. Bullseye Polytope in the white-box setting, achieves the attack success of 100% without our defense, but denoising with $\sigma$ of 0.25 reduces their success to 10–15%. We find the defense reduces the success of Poison Frog! and Convex Polytope from 79–95% to 3–15% at $\sigma = 0.1$. We also observe that the increase in $\sigma$ significantly degrades the model utility., e.g., $\sigma$ =1.0 loses the accuracy by 27%.

# D (DENOISED) POISONING SAMPLES

Table 8: **Visualize poisoning samples.** We, for the CIFAR10 training data, display the poisoning samples crafted by different clean-label poisoning attacks ($\ell_\infty$-norm of 16). We also show how the perturbations are denoised with difference $\sigma$ values in $\{0.1, 0.25, 0.5, 1.0\}$. $\sigma = 0.1$ yields to ineffective poisons. $\sigma = 0.0$ means we do not denoise the poisons.

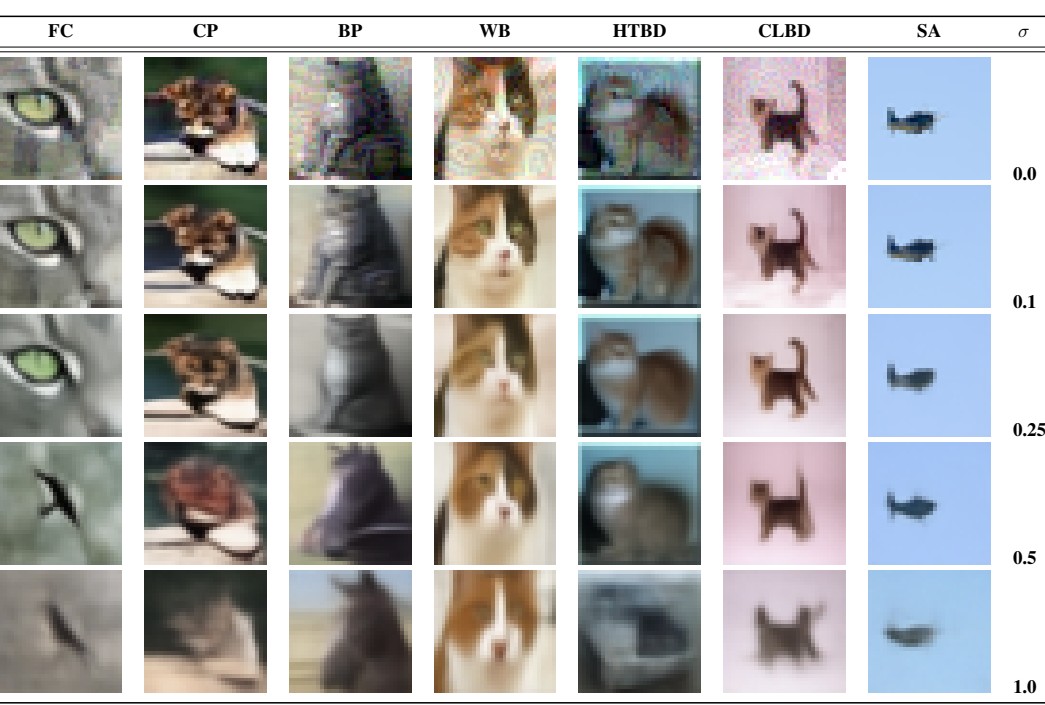

| FC | CP | BP | WB | HTBD | CLBD | SA | $\sigma$ |
|---|---|---|---|---|---|---|---|