# OpenReview forum: "Certified Robustness to Clean-label Poisoning Using Diffusion Denoising"
_ICLR.cc/2025/Conference — ICLR 2025 Conference Withdrawn Submission_

### Official Review · Reviewer_fM82 · 2024-10-16

**Soundness:** 2
**Presentation:** 2
**Contribution:** 2
**Rating:** 3
**Confidence:** 5

**Summary:**

This paper introduces a certified defense mechanism against clean-label poisoning attacks. The defense leverages a diffusion model, inspired by the adversarial robustness of randomized smoothing, to purify tampered training data. The proposed method is tested against seven different clean-label poisoning attacks, with minimal impact on test accuracy. Compared to existing countermeasures, this approach significantly lowers attack success while maintaining model performance.

**Strengths:**

1. The research approach is quite meaningful. Current defenses against poisoning attacks generally lack theoretical explanations, so employing a certified defense is a very promising idea.

2. The experimental results are excellent. The paper conducts defense experiments against both targeted poisoning attacks and backdoor attacks, and the results are very good.

**Weaknesses:**

1.The novelty of the approach is insufficient. In fact, there is already work [1] that proposes using the idea of diffusion denoising to defend against clean-label indiscriminate poisoning attacks, which reduces the novelty of this paper. The authors are encouraged to cite this related work for making this paper's contribution more clear. Moreover, the biggest issue with using diffusion purification to defend against poisoning attacks is whether the diffusion model was trained on a clean training dataset. If the diffusion model is trained on a clean training dataset (or even a small part), the entire defense approach becomes impractical. The paper should make this point more clearly, since I was wondering if the trained diffusion model had used clean training data.

2.Old backdoor attack baselines. The newest backdoor attack evaluated in this paper is 2022. The authors are supposed to evaluate more recent baselines [2].

3.In Line 437-438, the authors mentioned that "Ours and FrieNDs greatly reduce the attack success while preserving the model’s utility.The critical difference is that ours is a certified defense, while FrieNDs is not. A future work we leave is developing adaptive attacks against FrieNDs.", it indicates that the performance of the proposed diffusion denoising scheme is not the best. Therefore, I cannot find the absolute advantages of the proposed defense. The authors are supposed to explore more strengths of the proposed scheme to defeat existing baseline works.

4.Typos: Line 407: Table 5->Table 4. By the way, why does the Table 4 not include the results of Sleeper Agent?
Line 439-440 "Certified defenses are" ?

5.Why do the authors not report the defense results against Metapoison [3]?

[1]ECLIPSE: Expunging Clean-label Indiscriminate Poisons via Sparse Diffusion Purification. ESORICS'24

[2]Narcissus: A practical clean-label backdoor attack with limited information. CCS'23

[3]Metapoison: Practical general-purpose clean-label data poisoning. NeurIPS'20

**Questions:**

Please see the weakness.

---

### Official Review · Reviewer_VQLK · 2024-11-02

**Soundness:** 3
**Presentation:** 3
**Contribution:** 3
**Rating:** 5
**Confidence:** 3

**Summary:**

This paper proposes a novel method to evaluate certified robustness of clean-label poisoning attacks by applying diffusion denoising on the tampered training data. Testing the defense method against various clean-label attacks, the experiment results have shown that the proposed method can effectively reduce their attack success rate and outperform previous baselines in both robustness and model utility.

**Strengths:**

1. The paper is well written with clear descriptions and comprehensive experiments.
2. This is the first paper to apply the method of diffusion denoising to defense against clean-label poisoning attacks for certified robustness.
3. To evaluate the model, the author considers both model utility and robustness.

**Weaknesses:**

1. The method is hard to widely used for large scale. In Section 3.3, there is still a need to train 10k models for the certification process. With the scaling on both model and data scale, it is hard for it to be applied in real scenarios.
2. It would be better if there is a time cost comparison with other defense methods (both heuristic and certified defense), since performing the denoising process on the training set is still a huge cost.

**Questions:**

1. Why Figure 2 does not have the results for $\sigma=0.1, 0.5$? And Figure 1 and Figure 2 should be in the same figure for better computation.
2. From Table 5, I cannot see a clear improvement when comparing your method to ROE.

---

### Official Review · Reviewer_f7Fc · 2024-11-02

**Soundness:** 2
**Presentation:** 2
**Contribution:** 2
**Rating:** 5
**Confidence:** 5

**Summary:**

In this paper, the authors propose a certified defense against clean-label data poisoning attacks using diffusion denoising. The authors sanitize the training dataset by using an off-the-shelf diffusion model to remove adversarial perturbations on poisoned data. They also propose a warm-starting initialization to improve the model utility under the defense. Experiments are carried out on CIFAR-10, CIFAR-100 and Tiny ImageNet to evaluate the defense against multiple attacks and comparisons are performed with existing heuristic and certified defenses.

**Strengths:**

- The proposed approach is simple. The authors leverage the diffusion model to remove adversarial noise added on the poisoned data.

- The authors proposed the strategy of warm-starting to improve the model utility.

- The authors evaluate the proposed defense over difference settings such as training from scratch and transfer-learning.

**Weaknesses:**

I happened to be the review of this paper when it was submitted to ICLR2024 and NeurIPS this year. Compare to its prior version, the authors improved presentation and addressed some concerns, but still have the following issues:

- One of the most important claims of this work is that the proposed approach is a certified defense and it offers a provable guarantee. However, there is missing theoretical proof for the certified defense against poisoning attacks in the paper. The proposed defense is introduced follows the random smoothing, which is a certified defense against evasion attacks that is fundamentally different from poisoning attacks. The authors should clarify how the guaranteed robustness is achieved. In the prior rebuttal, the authors agreed to provide full proof in their Appendix, but  the proof is still missing in this version.

- I favor the simplicity of the proposed approach (i.e., the authors use an off-the-shelf diffusion model to remove the noise imposed on the poisoned samples), but the motivation of this idea is not well explained. Why pick diffusion model as a denoising approach over other techniques? Why does diffusion model work well in removing the adversarial perturbations?

**Questions:**

Please see weaknesses for details.

---

### Official Review · Reviewer_XPU2 · 2024-11-04

**Soundness:** 3
**Presentation:** 3
**Contribution:** 2
**Rating:** 5
**Confidence:** 4

**Summary:**

This paper introduces a defense mechanism against clean-label poisoning attacks by denoising training samples with a diffusion model prior to training. The goal is to mitigate the impact of poisoned samples, which might otherwise cause clean-label attacks.

**Strengths:**

1. By decoupling the training and defense processes, the proposed approach enables a training technique that enhances model accuracy while preserving robustness.
2. The authors provide extensive experimental evidence to support the effectiveness of their defense mechanism against clean-label poisoning attacks.

**Weaknesses:**

1. The paper claims to extend the algorithm by Cohen et al. (2019) to a clean-label poisoning setting. However, the method appears to be a straightforward extension of the work by Carlini et al. (2023) [1]. Comparing with the work proposed by Carlini, this work seems not sufficiently novel. Can the author(s) provide comparison of their proposed defense with the defense proposed by Carlini to claim their novelty?
2. In Table 1, the attack success rate is reported as nearly 99% without defense. This result suggests that the inclusion of a small fraction of poisoned data causes almost complete misclassification of all test samples of the target class $y_t$ to the clean label $y_b$ (correct me if I am wrong). Could the authors clarify how the models were trained? Line 283 states that the entire training dataset (50,000 samples for CIFAR-10) is used, implying that all 5,000 samples of the target class are included during training. If so, why do only 500 tampered samples cause such a high misclassification rate?
3. The difference between white-box and black-box attacks, mentioned in line 289, could be further elaborated to enhance the reader's understanding.
4. The writing could be improved for clarity and precision. Specific examples include:

    a) In line 83, clarification is needed regarding whether it should be "$y_b = y_p$" or "$y_b = y_p$.

    b) The optimization problem presented in line 85 is confusing. Terms such as $y_t$ and $y_{\text{adv}}$ are used interchangeably across lines 85, 88, and 91, which requires further clarification. Additionally, $y_{\text{adv}}$ should be explicitly defined—does it refer to the same label as $y_b$?

    c) Check the numbering of tables in lines 356 and 407.

    d) In Table 4, the terms TP and FP are used but never defined.




[1] Carlini, Nicholas, et al. "(certified!!) Adversarial robustness for free!." arXiv preprint arXiv:2206.10550 (2022).

**Questions:**

N/A

---

### Note · Authors · 2024-11-25

**Comment:**

We sincerely thank the reviewers for their time and effort in reading and evaluating our work. Due to some unforeseen family matters, the authors won't be able to complete the quality responses by this time and therefore decided to withdraw the submission. We hope we can address these aforementioned comments in the next submission with more fidelity.

**Withdrawal Confirmation:**

I have read and agree with the venue's withdrawal policy on behalf of myself and my co-authors.